# Immune Therapies in AL Amyloidosis—A Glimpse to the Future

**DOI:** 10.3390/cancers16081605

**Published:** 2024-04-22

**Authors:** Arnon Haran, Iuliana Vaxman, Moshe E. Gatt, Eyal Lebel

**Affiliations:** 1Department of Hematology, Hadassah Hebrew University Medical Center, Jerusalem 91120, Israel; arnon@hadassah.org.il (A.H.); rmoshg@hadassah.org.il (M.E.G.); 2Institute of Hematology, Davidoff Cancer Center, Rabin Medical Center, Petah Tikva 49100, Israel; juliava1@clalit.org.il

**Keywords:** plasma cell dyscrasia, amyloidosis, light chain, immunotherapy, bispecific antibodies, chimeric antigen receptor T cells

## Abstract

**Simple Summary:**

Light-chain (AL) amyloidosis is a rare disease similar to the more common disease, multiple myeloma (MM). Both are caused by proliferation of malignant plasma cells. In AL amyloidosis, disease is a result of the deposition of aggregates of proteins, namely immunoglobulin light chains, secreted by the malignant plasma cells, in target organs such as the heart or kidneys. Historically, treatment of AL amyloidosis has followed that of MM. A wide range of novel immunotherapies, i.e., therapies which utilize or activate immune mechanisms to eliminate the disease, are already established in MM and are gradually being adopted in AL amyloidosis as well. Although promising, the increased frailty of typical AL amyloidosis compared to MM patients is a concern in the administration of these therapies, which may be associated with severe side effects. We review both the promise and the challenges with the expansion of MM immunotherapies to AL amyloidosis.

**Abstract:**

Light-chain (AL) amyloidosis is a rare plasma cell disorder characterized by the deposition of misfolded immunoglobulin light chains in target organs, leading to multi-organ dysfunction. Treatment approaches have historically mirrored but lagged behind those of multiple myeloma (MM). Recent advancements in MM immunotherapy are gradually being evaluated and adopted in AL amyloidosis. This review explores the current state of immunotherapeutic strategies in AL amyloidosis, including monoclonal antibodies, antibody–drug conjugates, bispecific antibodies, and chimeric antigen receptor T-cell therapy. We discuss the unique challenges and prospects of these therapies in AL amyloidosis, including the exposure of frail AL amyloidosis patients to immune-mediated toxicities such as cytokine release syndrome (CRS) and immune effector-cell-associated neurotoxicity syndrome (ICANS), as well as their efficacy in promoting rapid and deep hematologic responses. Furthermore, we highlight the need for international initiatives and compassionate programs to provide access to these promising therapies and address critical unmet needs in AL amyloidosis management. Finally, we discuss future directions, including optimizing treatment sequencing and mitigating toxicities, to improve outcomes for AL amyloidosis patients.

## 1. Introduction

Light-chain (AL) amyloidosis is a rare plasma cell dyscrasia characterized by multi-organ damage due to misfolded immunoglobulin light chains (LC) secreted by clonal abnormal plasma cells (PC). In line with their common origin in clonal malignant PCs in the bone marrow, treatment of AL amyloidosis has traditionally followed that of multiple myeloma (MM), utilizing a combination of PC-targeted agents [1,2,3]. Recent development of therapies unique to AL amyloidosis, which aim to promote the clearance of the amyloidogenic LC from affected organs rather than targeting the abnormal PCs secreting the amyloidogenic LC, show great promise, yet are an adjunct to systemic anti-PC therapy [4]. Notwithstanding recently uncovered differences in underlying disease biology [5,6], treatment approaches successful in MM have effectively translated to treatment of AL amyloidosis. Thus, mirroring improvements in MM survival over the last decades with the introduction of novel therapeutic agents [7,8,9,10,11], progress has also been made in the treatment of AL amyloidosis [12]. Nevertheless, the prognosis of many patients, especially frail patients who are less able to tolerate the toxicity of effective therapies, remains dismal [13,14]. Although the concept of frailty in AL amyloidosis is not rigorously defined nor accompanied by disease-specific frailty scores, it is agreed that the main determinants of frailty and increased short-term mortality in these patients are older age and the extent of cardiac involvement at diagnosis. To a lesser degree, other organ involvement as well as pre-existing comorbidities also play a role in determining frailty [15]. Because improvement in organ function usually lags behind the cessation of further secretion of the toxic amyloidogenic LC [16], patients with advanced cardiac disease are not only less able to tolerate the side effects of treatment but also may succumb to pre-existing organ damage before gaining any benefit from therapy, as evinced by high rates of early mortality (within 3–6 months of diagnosis) in this patient population [14,17,18].

The aforementioned improvements in survival in both MM and AL amyloidosis in the last decades have largely been the result of widespread adoption of so-called novel agents, including proteasome inhibitors, immunomodulatory agents [10,19], and the anti-CD38 monoclonal antibody daratumumab, the first immunotherapy to gain the approval of the Food and Drug Administration (FDA) for treatment of MM [20] and subsequently AL amyloidosis [21]. More recently, the spectrum of immunotherapeutic agents (i.e., therapies that stimulate the host immune system to target cancerous cells) in MM has widened to include antibody–drug conjugates (ADCs), bispecific antibodies (BSAs), and chimeric antigen receptor T cells (CART). These are rapidly being integrated into the therapeutic armamentarium in MM, primarily in the context of relapsed/refractory disease [19], and are gradually being evaluated in AL amyloidosis as well.

Malignant PCs in MM and AL amyloidosis share several key features which are of relevance to immunotherapeutic approaches, including the surface expression of CD38 [22], B-cell maturation antigen (BCMA) [23], and G-protein-coupled receptor, class C group 5 member D (GPRC5D) [24]. These three proteins (Figure 1) are the targets of most immunotherapeutic agents developed to date [19] and, as referred to above, daratumumab, targeted against CD38, is already part of the current standard-of-care first-line treatment of AL amyloidosis, in combination with cyclophosphamide, bortezomib, and dexamethasone (CyBorD) [25].

As noted above, while, in MM, a host of immune therapies have already been approved and are rapidly becoming the mainstay of therapy, in AL amyloidosis, these approaches lag behind, with far less experience and unique challenges when compared with MM. Immune-mediated toxicities occur frequently as a result of T-cell activation with both BSAs and CART. These include cytokine release syndrome (CRS), a systemic inflammatory response characterized by fever, hypotension, hypoxia, and organ dysfunction, and immune effector-cell-associated neurotoxicity syndrome (ICANS), a toxic encephalopathy manifested by confusion, aphasia, somnolence, and, in severe cases, seizures and coma [26,27]. Development of CRS is a particular concern in cardiac AL amyloidosis patients, who often lack sufficient cardiovascular reserve to cope with this acute cardiocirculatory stress [28,29,30,31]. Notwithstanding such safety concerns, immunotherapy holds special promise in AL amyloidosis from the standpoint of efficacy. While direct comparison with more traditional therapies is difficult in the absence of head-to-head trials, high rates of rapid and deep responses are seen with novel immunotherapies, even in the r/r setting in which they have been evaluated to date [32]. The achievement of a rapid and deep hematologic response is one of the main goals of treatment in AL amyloidosis, preventing further end-organ damage and resulting in improved survival [33,34]. Whereas depth of response and, in particular, the achievement of minimal residual disease (MRD) negativity predicts survival in MM [35,36], the prognostic impact of response kinetics (i.e., the rapidity of the decline in the serum paraprotein following initiation of therapy) is controversial, with very early response paradoxically associated with worse long-term outcomes [37,38]. In contrast, rapidity of response is crucial in ensuring optimal outcomes in amyloidosis [39]. With respect to toxicity, although hardly devoid of side effects in most cases, immunotherapies offer a side effect profile unique from that of current conventional treatments and one which may prove to be better tolerated by AL amyloidosis patients.

In the following, we review the experience to date with the use of PC-directed immunotherapies in AL amyloidosis. The above-mentioned therapies promoting clearance of amyloid fibrils will not be discussed here.

## 2. Monoclonal Antibodies

Monoclonal antibodies directed against CD38 and, to a lesser degree, signaling lymphocytic activation molecule-F7 (SLAMF7) have proven to be of value in the treatment of MM. Two anti-CD38 antibodies, daratumumab and isatuximab, have been approved by the Food and Drug Administration (FDA) for the treatment of MM. Although pharmacologically similar, the two agents differ in their method of administration, with daratumumab being administered subcutaneously and isatuximab available only as an intravenous formulation, as well as in the CD38 epitope targeted and in some of the mechanisms by which they promote cell death. Nevertheless, patients refractory to one agent usually will not respond to the other [40]. Daratumumab was incorporated into standard first-line therapy in AL amyloidosis following the results of the ANDROMEDA trial in 2021. In this trial, 388 patients with newly diagnosed AL were randomized to receive treatment with a standard induction regimen of bortezomib, cyclophosphamide, and dexamethasone, either alone (CyBorD) or with daratumumab (D-CyBorD) [41]. The addition of daratumumab resulted in a nearly three-fold increase in the rate of hematologic complete response (CR), the primary endpoint of the trial, which was achieved in 53% of patients receiving daratumumab as compared with 18% of patients in the control group. The rate of target organ response, including both cardiac and renal, was approximately doubled with the addition of daratumumab. The trial also highlighted a relatively favorable safety profile for D-CyBorD, with only a slight increase in hematologic and infectious adverse events when compared to CyBorD, and a nearly identical rate of treatment discontinuation due to adverse events. Patients with severe cardiovascular disease, including an N-terminal pro-B-type natriuretic peptide (NT-proBNP) level of more than 8500 ng/L, a systolic blood pressure of less than 90 mmHg, or a New York Heart Association (NYHA) classification of stage IIIB or IV at screening, were excluded from the trial. This study did not include an endpoint of bone marrow MRD negativity, which may become of high importance, since a major aim is to eradicate the toxic amyloidogenic light chain that is causing the end-organ damage [33].

Daratumumab is currently being evaluated both in patients with more advanced cardiac involvement at diagnosis who were not included in initial trials and as part of alternative combination therapies, especially in the r/r setting (Table 1). Patients with advanced cardiac disease have historically had a grave prognosis, with a median overall survival of 7 months in patients with Mayo stage 3 disease and just 4 months in patients with stage 3b disease [42]. Although early mortality rates in the ANDROMEDA trial were not reduced between groups [43], it is nevertheless hoped that achieving a deep response as fast as possible with minimal toxicity may overcome these high mortality rates. As monotherapy in newly diagnosed patients, daratumumab is being assessed in a phase 2 study (NCT04131309) in patients with Mayo cardiac stage 3b whom, as noted above, were intentionally excluded from the ANDROMDA trial. Bortezomib and dexamethasone may be added in patients with a very good partial response (VGPR) or better after three cycles of treatment. Early results from this trial have been favorable [44], showing rapid and deep hematological responses and an overall survival rate of 65% at 6 months, higher than reported with previous therapies in this patient group. A study evaluating the combination of daratumumab, bortezomib, and dexamethasone in newly diagnosed patients with stage 3a/3b disease is also currently ongoing (NCT04474938). Early results [45] have noted high hematological response rates but also a high rate of early mortality, 15% at 1 month and 17.5% at 3 months; whether daratumumab monotherapy or sequential treatment will lead to lower early mortality rates remains to be seen. The duration of treatment with daratumumab is controversial and the effect on event-free survival of daratumumab maintenance therapy following successful induction with D-CyBorD is being studied in a phase 2 trial (NCT05898646).

To date, a large number of retrospective studies have reported on the use of daratumumab in the r/r setting [43,46], and daratumumab as monotherapy in previously treated disease was prospectively evaluated in two phase 2 clinical trials, with relatively high rates (55–86%) of rapid and deep (VGPR or better) hematological responses [47,48]. Following the success of daratumumab monotherapy in r/r disease, various combination treatments are currently being prospectively tested, including daratumumab with pomalidomide (NCT04270175 and NCT04895917) or ixazomib (NCT03283917) and daratumumab with venetoclax and dexamethasone (NCT05486481) in patients with the 11;14 translocation.

Several trials of isatuximab in AL amyloidosis are ongoing. In the upfront setting, isatuximab is being evaluated in combination with bortezomib, cyclophosphamide, and dexamethasone in patients with high-risk disease (NCT04754945). This study includes patients with advanced cardiac disease, including Mayo stages 3 and 4, and utilizes what has been termed a “slow-go” approach, comprising the sequential addition of therapies and gradual dose escalations to limit treatment-related toxicity and early mortality. In the r/r setting, trials are evaluating both isatuximab monotherapy (NCT03499808) and the combination of isatuximab, pomalidomide, and dexamethasone (NCT05066607).

Elotuzumab is a monoclonal antibody targeting the surface antigen SLAMF7 and is approved for the treatment of patients with r/r MM in combination with immunomodulatory agents and proteasome inhibitors. Successful treatment with a combination of elotuzumab, lenalidomide, and dexamethasone was described in a single case report of a patient with concurrent MM and AL with renal involvement. Despite the patient being heavily pretreated, this regimen resulted in a rapid and prolonged hematologic and organ response, with over 2 years of follow-up reported [49]. The combination of elotuzumab, lenalidomide, cyclophosphamide, and dexamethasone is being evaluated in a phase 2 clinical trial in patients with r/r disease (NCT03252600).

## 3. Antibody–Drug Conjugates

Belantamab mafodotin (Bela) is an antibody–drug conjugate (ADC) consisting of the anti-BCMA monoclonal antibody belantamab conjugated to monomethyl auristatin F (MMAF), a tubulin inhibitor. Bela was approved by the FDA in 2020 as a single agent for the treatment of r/r MM based on the results of the phase 2 DREAMM-2 study [50]. Approval was subsequently withdrawn when a confirmatory phase 3 study failed to show benefit in progression-free survival when compared to pomalidomide and dexamethasone [51]. A prospective study investigating Bela in r/r AL amyloidosis, at a dose of 2.5 mg/kg every 6 weeks, is currently ongoing (NCT04617925). A preliminary report after enrollment of 25 out of a planned 36 patients, with a median three prior lines of therapy, showed a high overall hematologic response rate (partial response or better) of 60% and a 3-month organ response rate of 20%; responses were rapid, with a median time to first hematologic response of 15 days. However, at cutoff and with a median follow-up of 11 months, only six (24%) patients remained on the study treatment, with nine patients having discontinued treatment due to disease progression and seven due to adverse events. Ocular adverse events occurred in practically all patients, including four patients with grade 3 keratopathy [52]. Other data come from small retrospective series. Zhang et al. reported on six r/r AL amyloidosis patients with 5–10 prior lines of therapy receiving Bela at the standard dose of 2.5 mg/kg every 3 weeks, with five out of six patients achieving hematological response and four out of five patients with cardiac involvement achieving cardiac response. Similar to MM, keratopathy was the most frequent adverse event but limited to grade 1-2 in this series [53]. Khwaja et al. described 11 r/r AL patients with 2–4 prior lines of therapy who received Bela, with an overall and complete response rate of 64% and 40%, respectively. Keratopathy occurred in 8 out of 11 patients, requiring dose or schedule modification in four patients and treatment cessation in one patient. Overall, most patients required dose reduction either at initiation or during therapy, with only one patient receiving the standard dose of 2.5 mg/kg every 3 weeks for more than three cycles of treatment; high response rates were nevertheless observed [54]. Finally, together with our Italian colleagues, we summarized the experience with 12 r/r AL patients from five sites (manuscript under review). In line with the above data, encouraging efficacy was observed, with an overall hematologic response rate of 75%, organ response in 50% of patients, and a median duration of response of 34 months. Again, keratopathy was frequent (83%) but was overall manageable with appropriate dose interruptions and reductions, and no AL-involved organ deterioration was noted.

Interestingly, response rates of 60–80% seen in r/r AL amyloidosis are approximately twice those seen in r/r MM in similarly heavily pretreated patients [50,55]. This difference may be a result of the lower burden of disease (in terms of plasma cell clone size) in AL amyloidosis compared to MM and may allow even greater dose reductions than those reported above so as to minimize ocular toxicity while still maintaining high efficacy. Besides the apparently greater efficacy, the data from the small case series above also indicate lower rates of hematological toxicity in AL amyloidosis compared to MM, also likely related to reduced marrow involvement and better marrow reserve in AL amyloidosis patients. In summary, limited data suggest that Bela may be a valuable agent in AL amyloidosis, with encouraging efficacy and acceptable toxicity in this frail population.

STI-6129 is a novel ADC consisting of an anti-CD38 antibody also linked to MMAF and is currently being evaluated in a phase 1/2 clinical trial (NCT04316442).

## 4. Bispecific Antibodies

BSAs are antibodies recognizing two different epitopes, most commonly CD3 and a tumor epitope, to facilitate T cell redirection and activation, ultimately resulting in T-cell-mediated target cell killing [56]. BSAs have shown impressive results in r/r MM, although high rates of infectious complications remain a concern. Teclistamab is a BCMA-CD3 BSA and was the first BSA approved by the FDA for the treatment of r/r MM following the results of the phase 1–2 MajesTEC-1 trial [57]. Published experience in AL amyloidosis, consisting of one case report and two small case series, shows high rates of rapid and deep hematologic responses and allays some of the theoretical concerns regarding the development of CRS in these frail patients. The case report [58] described an AL patient with cardiac, renal, and soft tissue involvement in the setting of relapsed MM with six prior lines of therapy. Response after six cycles of treatment with teclistamab was favorable, with a reduction in NT-proBNP levels and proteinuria, and there were no adverse events reported. A case series describing 17 patients also provided encouraging results [59]. All but one patient in this series had cardiac involvement, and 10/17 (59%) had Mayo stage 3 disease. In this heavily pretreated cohort, with a median of four prior lines of therapy, VGPR or better was achieved in 88% of patients, and 76% of patients had a decrease in the involved free light chain (FLC) below 10 mg/L. Importantly, responses were rapid, with a median time to best hematologic response of one month. Median follow-up at the time of publication was short (3 months), but five patients (29%) had already achieved an organ response. Unfortunately, high rates of adverse events were also seen, including five patients (29%) with severe infections, including one death and one case of severe ICANS requiring treatment discontinuation. These rates are, however, consistent with those seen in MM, and no additional signals relating to cardiac or kidney toxicity specific to AL patients were identified. Notably, although nine (53%) patients developed CRS, all cases were grade 1. A second case series, describing seven patients with r/r AL amyloidosis, most with concurrent MM and a median six lines of prior therapy, showed similarly impressive results [60]. All patients achieved partial response or better, and six of seven patients achieved a reduction in the difference between involved and uninvolved light chain (dFLC) to less than 1 mg/dL at one month of treatment. Organ response was seen in three of four patients with cardiac involvement. Rates of CRS were consistent with previous reports, occurring in four out of seven patients (57%), but all cases were grade 1 and there were no cases of ICANS. Two patients developed grade 3 or higher infectious complications. Thus, although limited in size, these reports provide reassurance with regards to safety concerns specific to AL amyloidosis in general and cardiac disease in particular, while also showing high efficacy as expected from previous experience with teclistamab in r/r MM. Experience in AL with other BSAs, such as talquetamab and elranaramab, has yet to be reported in the literature.

## 5. Chimeric Antigen Receptor T-Cell Therapy

CART has emerged as an efficient immune therapy in r/r MM [61]. This unique modality involves the collection of autologous T lymphocytes from the patient and the manipulation of these cells to enable both the recognition of a tumor antigen and the consequent activation of the T cells, resulting in the elimination of the tumor cells. While BCMA is the main target antigen in MM tested to date, encouraging data on other targets and approaches are gradually accumulating [62,63].

Two anti-BCMA CART products, idecabtagene vicleucel (ide-cel) and ciltacabtagene autoleucel (cilta-cel), are approved by the FDA and European Medicines Agency (EMA) for the treatment of r/r MM. Although currently approved only after four or more previous lines of therapy, clinical trials are investigating the use of CART in earlier lines of treatment [64]. Recently, two phase 3 randomized clinical trials in either the second-line (cilta-cel) or third-line (ide-cel) setting showed improved progression-free survival with single infusions of CART products when compared to various standard-of-care treatments [65,66]. Thus, expansion to earlier lines of therapy in MM, including as a possible alternative to autologous stem cell transplant in the frontline setting, is expected in the near future [64]. Several drawbacks remain, including both early and late unique adverse events and limited availability due to the complexity of the manufacturing process [67].

Late toxicities include the development of prolonged cytopenias and hypogammaglobulinemia, with a resulting increased risk of infections [68]. Less commonly, secondary cancers have been reported, including 22 cases of T-cell lymphoma arising directly from the CART product [69] and the occurrence of myeloid neoplasms (myelodysplastic syndrome or acute leukemia) in as many as 10% of patients in some studies [70,71]. The mechanism underlying the latter toxicity remains unclear.

Early toxicity, as mentioned above, is a result of T-cell activation and cytokine release and includes the development of CRS, ICANS, and hemophagocytic lymphohistiocytosis or macrophage activation syndrome, usually within the first 1 to 3 weeks following CART cell infusion. Accumulating experience has led to strategies to either prevent or more effectively manage these complications to avoid their most severe sequelae, using a combination of corticosteroids and anti-IL-6 therapy, most commonly tocilizumab [72]. Of direct relevance to patients with AL amyloidosis are the cardiovascular adverse events associated with CART therapy, usually as part of the spectrum of organ damage accompanying the development of severe CRS and ranging from an elevated troponin to overt heart failure seen even in patients with normal baseline cardiac function [28,31]. Trepidation with exposure of already frail AL amyloidosis patients to these adverse events, along with the relative rarity of this disease, probably underlie the fact that commercial CART products have not been tested in AL amyloidosis thus far. Indeed, AL amyloidosis was defined as an exclusion criterion in both of the phase 3 studies mentioned above [65,66]. However, several locally developed academic CART products have been evaluated in these patients.

Oliver-Caldes and colleagues presented the first case of a patient with AL amyloidosis successfully treated with BCMA-targeted CART therapy [73]. The patients, who developed amyloidosis with renal involvement during the course of MM, was treated with CART with tolerable acute toxicity comprising grade 1 CRS, transient grade 4 cytopenias, and infectious complications, including SARS-CoV2 pneumonia and BK virus hemorrhagic cystitis. Hematological stringent complete response and MRD negativity were achieved by 3 months after CART cell infusion. Organ response manifested in a 55% decrease in 24 h urinary protein excretion by 6 months and 70% decrease at 12 months post-treatment.

Das et al. reported on the use of commercial CART products in two patients with AL amyloidosis, concurrent r/r MM, and renal and cardiac involvement, one a 62-year-old woman with Mayo stage 2 disease treated with ide-cel and the other a 33-year-old male with Mayo stage 4 disease treated with cilta-cel. Both patients received prophylactic dexamethasone, with the former developing no signs or symptoms of CRS and the latter developing grade 3 CRS requiring treatment with tocilizumab and admission to an intensive care unit for vasopressor support but resolving within 24 h. Both hematologic response and cardiac response, as evidenced by a >30% decrease in NT-proBNP levels at 9 months post-treatment, were achieved in both patients [74].

We recently reported on our experience with HBI0101/NXC-201, a locally developed and produced academic anti-BCMA CART with a 4-1BB/CD137 co-stimulatory domain. A manufacturing time of only 10 days allows rapid treatment, obviating the need for bridging therapy. Uniquely, the phase 1 and 2 studies of HBI0101/NXC-201 were designed to include, in addition to a preponderance of MM patients, both patients with AL amyloidosis, including those with cardiac involvement and patients with advanced renal failure (stage 3b or higher chronic kidney disease), who were excluded from trials of commercial CART products. Clinical studies evaluating HBI0101/NXC-201 began February 2021 with a phase 1a dose-escalation study, starting with the infusion of 150 × 10^6^ CAR-positive T cells and proceeding to 450 × 10^6^ and to the target dose of 800 × 10^6^ cells. A standard conditioning regimen of fludarabine and cyclophosphamide was given as lymphodepletion prior to CAR-T therapy in patients with adequate renal function, while bendamustine was used as an alternative but equally effective regimen [75] in patients with a creatinine clearance of less than 30 mL/min. Following the encouraging results of the phase 1a trial, from the points of view of both efficacy and toxicity [76], the target dose of 800×10^6^ cells was further assessed in phase 1b and 2 studies, including over 80 patients (manuscript under revision).

Despite lower expression of BCMA in AL PCs when compared to MM controls, ex vivo validation studies confirmed efficient targeting of these cells by HBI0101/NXC-201 CART cells, and an initial 4 [77] and subsequently a total of 11 AL amyloidosis patients out of a target 40 have been treated to date with HBI0101/NXC-201 CART. Results from eight of these patients were previously reported [77,78].

Of the 11 patients, 9 were treated within the clinical study and 2 on a compassionate care basis due to concomitant myelodysplasia and low performance status, which precluded enrollment in the trial. One patient received 150 × 10^6^ CART cells, two patients received 450 × 10^6^ cells, and eight patients received 800 × 10^6^ cells. Median age in this cohort was 64. Two patients had concomitant MM, and nine out of eleven had cardiac involvement, including four with Mayo stage 3 disease. Patients were heavily pretreated, with a median six prior lines of therapy, all were triple refractory (refractory to a proteasome inhibitor, an immune modulator drug, and an anti-CD38 antibody), and most (6/11, 55%) were refractory to Bela.

CART therapy was found to be relatively safe in these patients, with 9 out of 11 patients experiencing CRS but only 2 cases of grade 3 CRS and no cases of Grade 4 or 5 CRS. Patients with stage 3b cardiac disease were admitted to the cardiac intensive care unit prior to cell infusion in anticipation of possible cardiac decompensation. Otherwise, standard supportive and therapeutic measures were used, including administration of corticosteroids and tocilizumab as needed. There were no cases of ICANS and no treatment-related deaths. Cytopenias were less severe than those seen in MM patients and usually resolved early. Transient peri-infusional worsening of heart or kidney function were in all cases manageable with standard supportive care and were reversible.

Striking efficacy was observed. In 10 of 11 evaluable patients, the overall hematologic response rate was 100%, with 7 CRs, 2 VGPRs, and 1 partial response. At day 30 post-CART infusion, six of nine evaluable patients were MRD-negative at 10^−5^ by multicolor flow cytometry. Duration of response was variable, ranging between 1.5 months and ongoing after 22 months. Early responses translated to organ responses in five out of nine patients with target organ involvement, including cardiac, renal, and hepatic responses. However, five patients, including three with progressive disease, one who was in VGPR, and one who was in PR, succumbed to cardiac disease within the first year after CART infusion, and another patient died due to SARS-CoV2 pneumonia while in CR.

The above case reports and our experience with HBI0101/NXC-201 anti-BCMA CART therapy in the context of a prospective clinical trial provide proof of concept that CART is both safe and highly efficacious for the treatment of AL. Due to the deep and quick reduction in light-chain secretion, organ response can be observed early. In advanced cardiac amyloidosis patients, though there were no deaths directly related to treatment toxicity, cardiac-related deaths in the first year (most while in hematologic response) appear frequent, suggesting that use of CART earlier in the disease may provide better organ responses and survival.

## 6. Conclusions and Future Directions

In the near future, it is expected that anti-PC immune therapies that have proved efficacious in MM will be further tested in AL, particularly CART and BSA. While showing a manageable toxicity profile for frail AL patients, even with extensive cardiac involvement and multiple previous lines of therapy, they promise unprecedented rates of hematological responses, which are rapid and deep, including MRD negativity and low dFLCs, thus translating to swift organ responses. Furthermore, it is likely that usage of these therapies early in the course of disease will maximize their favorable benefit. A critical unmet need in AL is the management of patients who achieved suboptimal hematologic responses (did not achieve CR after induction), as the depth of hematological response is crucial in AL amyloidosis. Hence, international initiatives to conduct prospective clinical studies with these therapies in AL should be strongly advocated, as well as compassionate programs to allow AL patients to obtain access to promising therapies in the meanwhile. Moreover, efforts should be made to understand the optimal sequencing of these therapies in AL patients and mitigate toxicities. Studies evaluating the use of BSA in a fixed-duration schedule accompanied by robust infection prophylaxis measures are encouraged. In parallel to efforts towards more effective anti-PC therapies, therapies aiming to interrupt the deposition of amyloid fibrils, prevent organ damage, and promote amyloid fibril clearance, although outside the scope of the current review, are also showing promising results in advanced clinical trials [4] and are expected to become a powerful adjunct to anti-PC therapies in the near future.

## Figures and Tables

**Figure 1 cancers-16-01605-f001:**
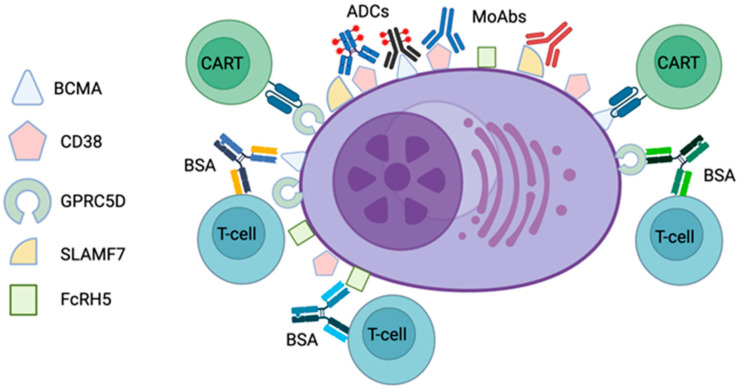
Schematic representation of plasma cell surface antigens targeted by current immunotherapeutic approaches in multiple myeloma and light-chain amyloidosis. BCMA—B-cell maturation antigen; GPRC5D—G-protein-coupled receptor, class C group 5 member D; SLAMF7—signaling lymphocytic activation molecule, F7; FcRH5—Fc receptor-homolog 5; ADC—antibody–drug conjugate; MoAb—monoclonal antibody; BSA—bispecific antibody; CART—chimeric antigen receptor T cell. Created with the help of BioRender.

**Table 1 cancers-16-01605-t001:** Ongoing and future clinical trials of immunotherapy in light-chain amyloidosis; R/R—relapsed/refractory. Immune therapies are marked in bold in the intervention column.

Treatment Type	Intervention	Phase	Disease Setting	Clinicaltrials.gov Identifier	Start Year	Completion Year	Target Patient Number
Monoclonal antibodies	**Daratumumab** monotherapy	2	Newly diagnosed, stage 3b	NCT04131309	2023	2025	40
	**Daratumumab**, pomalidomide, dexamethasone	2	R/R	NCT04270175	2021	2024	21
	**Daratumumab**, bortezomib, dexamethasone	2	Newly diagnosed, stage 3	NCT04474938	2021	2023	40
	Venetoclax, **daratumumab**, dexamethasone	1–2	R/R, t(11;14)	NCT05486481	2023	2027	78
	**Daratumumab**, ixazomib, dexamethasone	1	Newly diagnosed and R/R	NCT03283917	2018	2025	21
	**Daratumumab** monotherapy	2	Maintenance	NCT05898646	2023	2024	96
	**Daratumumab**, pomalidomide	2	R/R	NCT04895917	2021	2024	40
	**Isatuximab** monotherapy	2	R/R	NCT03499808	2018	2023	43
	**Isatuximab**, pomalidomide, dexamethasone	2	R/R	NCT05066607	2022	2025	46
	**Istauximab**, bortezomib, dexamethasone, cyclophosphamide	1	Newly diagnosed	NCT04754945	2021	2025	25
	**Elotuzumab**, lenalidomide, dexamethasone, cyclophosphamide	2	R/R	NCT03252600	2017	2023	53
Antibody-drug conjugates	**STI-6129** (antibody-drug conjugate)	1b/2a	R/R	NCT04316442	2021	2024	60
	**Belantamab mafodotin**	1/2a	R/R	NCT05145816	2023	2026	37
	**Belantamab mafodotin**	2	R/R	NCT04617925	2021	2025	35
CAR-T cells	**FKC288**	1	R/R	NCT05978661	2023	2025	12
	**HBI0101/NXC-201**	1b	R/R	NCT06097832	2024	2025	40

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
