# Peer review of "Immune Therapies in AL Amyloidosis—A Glimpse to the Future"

_cancers, 2024, doi:10.3390/cancers16081605_

Round 1
Reviewer 1 Report
Comments and Suggestions for Authors
Dear authors,
The review is well-written and extensive. I have a few minor comments.
1. The acronym r/r is first used on line 98, however, its explanation is given in the captions to the Table 1 on line 188. Please, correct it. Acronyms should be explained on the first citation.
2. Please, explain, what is the difference between daratumumab and isatuximab (first mentioned on line 117).
3. I'm confused about the data on CART treatments provided in Table 1 (2 trials with 12 and 40 patients) and in the text (line 318 - 1 patient, line 327 - 2 patients, line 355 - 4-11 patients). Please, clarify this situation.
Author Response
Dear reviewer,
Thank you for taking the time to read and improve our manuscript.
With regards to the comments offered:
- We have added the acronym explanation to its first occurrence in line 68 of the introduction.
- We added the following with regards to differences between isatuixmab and daratumumab: "Two anti-CD38 antibodies, daratumumab and isatuximab, have been approved by the Food and Drug Administration (FDA) for the treatment of MM. Although pharmacologically similar, the two agents differ in their method of administration, with daratumumab being administered subcutaneously and isatuximab available only as an intravenous formulation, as well as in the CD38 epitope targeted and in some of the mechanisms by which they promote cell death. Nevertheless, patients refractory to one agent usually will not respond to the other [40]. "
- The patient numbers in table 1 refer to target patient numbers at the conclusion of the ongoing clinical trials. We have added to the section regarding the HBI0101 trial to clarify that the 11 patients treated to date are part of the target of 40.
Thank you again for your time and we hope that the above changes made are satisfactory.
Sincerely and on behalf of all authors,
Arnon
Reviewer 2 Report
Comments and Suggestions for Authors
The review entitled: “Immune therapies in AL amyloidosis – a glimpse to the future” (ID: cancers-2950479) by Haran et al. aims to review an update with the use of PC-directed immunotherapies in AL amyloidosis.
Albeit the review is well written and of special interest, minor comments should be addressed to further improve the manuscript.
Comments:
1. In the manuscript “frail” should be more defined.
2. Future directions: The authors should more define how the deposition of amyloid fibrils and the prevention of organ damage and their clearance could be encouraged in the future.
Author Response
Dear reviewer,
Thank you for taking the time to read and improve our manuscript.
With regards to the comments offered:
- We have added the following definition of frailty to the introduction:"Although the concept of frailty in AL amyloidosis is not rigorously defined nor accompanied by disease-specific frailty scores, it is agreed that the main determinants of frailty and increased short-term mortality in these patients are older age and the extent of cardiac involvement at diagnosis. To a lesser degree, other organ involvement as well as pre-existing comorbidities also play a role in determining frailty [15]. "
- We have altered this sentence to better reflect its intended meaning: "
In parallel to efforts towards more effective anti PC therapies, therapies aiming to interrupt the deposition of amyloid fibrils, prevent organ damage, and promote amyloid fibril clearance, although outside the scope of the current review, are also showing promising results in advanced clinical trials, and are expected to become a powerful adjunct to anti PC therapies in the near future [4]."
Thank you again for your time and we hope that the above changes made are satisfactory.
Sincerely and on behalf of all authors,
Arnon